# Quantum Elastica

**DOI:** 10.3390/e27040388

**Published:** 2025-04-06

**Authors:** Davi Geiger, Michael Werman

**Affiliations:** 1Department of Computer Science, Courant Institute of Mathematical Sciences, New York University, New York, NY 10012, USA; 2Institute of Computer Science, The Hebrew University, Jerusalem 9190401, Israel; michael.werman@mail.huji.ac.il

**Keywords:** elastica, quantum theory, quantum path integral, statistics, information

## Abstract

This paper presents a quantum method to tackle optimization challenges. Departing from the typical applications of quantum theory in particle physics, we demonstrate our approach using the elastica problem as a concrete example. The elastica, a classic variational problem extensively studied by mathematicians, serves as an ideal test case. Within quantum theory, our central innovation lies in the way we handle boundary conditions by combining forward and backward propagating wave solutions, a concept inspired by the superposition of forward and backward time-traveling particle waves in quantum mechanics. This approach not only provides a novel solution method for the elastica problem but also opens new pathways for applying quantum mathematical techniques to classical optimization challenges in other domains.

## 1. Introduction

This paper introduces a novel quantum-inspired method to solve optimization problems, venturing beyond the typical applications of quantum theory in particle physics. We demonstrate our approach using the elastica problem, which describes the shape of bent elastic rods, as a concrete example. A key innovation lies in our treatment of boundary conditions within a quantum framework: we achieve this by combining forward and backward propagating wave solutions. This is conceptually inspired by the combination of forward and backward time-traveling particle waves in quantum mechanics (QM). Our approach differs from Wheeler–Feynman absorber theory [1] as we employ multiplication and summation of the forward and backward solutions, hinting at potential implications for future physical theories.

This work builds on a rich history of physics-inspired computation. Kirkpatrick’s Simulated Annealing method [2] transformed optimization by borrowing concepts from statistical mechanics, and Hopfield [3,4] innovated in neural networks using insights from physical systems. We extend this tradition into the quantum theory domain. Although quantum computing has already demonstrated the power of quantum principles in computation through algorithms such as Shor’s factorization [5], our approach shows that quantum mathematics, even without quantum hardware, can offer novel solutions and insights regarding classical optimization challenges. The paper presents the theoretical underpinnings of the method; computational and complexity issues will be addressed in a follow-up paper.

### 1.1. Quantum Method as a (New) Probabilistic Method

We start by highlighting fundamental distinctions between the quantum method and classical probabilistic approaches. Dirac [6] pointed out a key difference: the concept of state superposition leading to interference. Additionally, entanglement and the indistinguishability of multiple elementary objects are unique to quantum probabilities. Another distinction in quantum theory arises from the nonlinearity of quantum probability dynamics. Although the amplitude wave evolves linearly according to the quantum kernel, the associated quantum probability follows a nonlinear trajectory, governed by the application of Born’s rule to the wave function.

Quantum algorithms, algorithms that run on a realistic model of quantum computation, have been proposed [5,7,8,9]. They leverage essential features of quantum interference and entanglement. These algorithms are compared to classical algorithms based on parameters such as computational speedup and quantum resource requirements (qubits). Studies have shown that classically simulating a quantum algorithm for certain problems, such as linear equations in certain parameter regimes, can outperform classical algorithms [10,11]. These algorithms suggest that quantum theory can be applied to non-physical problems.

Quantum theory offers multiple approaches to construct probabilistic methods from classical theories. For instance, geometric quantization leverages Dirac’s concept of transforming classical Poisson brackets into non-commutative operators [12]. The approach we employ here is Feynman’s path integral formulation [13,14], which necessitates a Lagrangian, i.e., the optimization of the integration of the Lagrangian, an action that is directly suited to many optimization problems.

We view quantum mechanics as essentially a quantum method applied to Lagrangian mechanics, and, similarly, quantum elastica represents the application of this same quantum method to the Lagrangian formulation of the elastica problem. We contend that this approach generalizes so that any problem expressible using a Lagrangian (“Lagrangian-X”) can be addressed using an analogous quantum method, potentially leading to more robust and precise solutions. Our approach draws upon Feynman’s work [14], demonstrating an algebraic structure within quantum probabilities; classical logic operations **or** and **and**, corresponding to addition and multiplication in classical probability, are extended to the addition and multiplication of quantum states or wavefunctions. While some of these operations are well established in physics, others, which we employ for the elastica problem, represent novel applications within the field. More specifically, and as mentioned above, a key innovation here is our treatment of boundary conditions within a quantum framework by combining forward and backward propagating wave solutions.

### 1.2. Possible Advantages of the Quantum Method

A natural question to ask is why employ a quantum method and not a classical statistical method to solve an optimization problem? Here, we outline some of the advantages that the quantum method offers in physics, and we only speculate on what they may provide to other domains.

The initial success of quantum mechanics lay in its ability to accurately predict atomic-level phenomena, providing fundamental insights into the stability of matter. Interference effects, specifically, enable the cancellation of probabilities, leading to the quantization of energy levels and stable orbital states. Feynman’s analysis of the double-slit experiment [14] demonstrates this phenomenon: interference not only generates a wave pattern, but also enhances the maximum probability amplitude while reducing neighboring probabilities. This results in a sharper, more isolated maximum, a more precisely defined peak compared to the classical predictions.

Previous work has demonstrated the effectiveness of complex-valued kernels in detecting triangles and circles, employing these kernels to model probabilistic errors in parameter space [15,16,17]. This research showed that the inherent cancellation properties of complex kernels yield results that are significantly more robust to noise and clutter than classical methods.

We hope that a quantum-inspired approach provides a more robust and accurate solution to classical optimization problems than traditional statistical methods. However, we clarify that this paper is about showing the possibility of employing the quantum method in other domains, and some novelties we found when applying it to the elastica problem. However, we leave it for future work to demonstrate with empirical studies and further analysis if it is indeed better to do so.

### 1.3. The Elastica Curve

Let a 2D curve $\Gamma_{L}$ of length *L* be parameterized by arc length *s*, where $\frac{dx(s)}{ds}=\left(\cos\theta (s),\;\sin\theta (s)\right)$ has unit velocity and θ(s) is the tangent angle of the curve. Given a starting position and orientation $\left\{x^{0},\theta^{0}\right\}$ and an end position and orientation $\left\{x^{L},\theta^{L}\right\}$ start–end conditions, the elastica curve $\Gamma_{L}^{*}$ minimizes(1)$$S\left(\Gamma_{L}\right)=\int_{\Gamma_{L}}\;\kappa^{2}\left(s\right)\;ds\;,$$
where κ(s) is curvature.

We refer to the minimization criterion $S(\Gamma_{L})$ as the action of the elastica, the integral of the Lagrangian $L\left(\theta \left(s\right),\dot{\theta }\left(s\right),s\right)={\dot{\theta }}^{2}\left(s\right)$, where $\dot{\theta }\left(s\right)=\frac{d\theta (s)}{ds}=\kappa \left(s\right)$.

The elastica Lagrangian has some similarities to the free particle Lagrangian in classical physics, where the arc length, *s*, replaces time, *t*, the tangent θ(s) replaces the particle position y(t), and the kinetic energy $\frac{1}{2m}{\left|\frac{dy(t)}{dt}\right|}^{2}$ is replaced by ${\left|\frac{d\theta (s)}{ds}\right|}^{2}$. However, there are important differences: (i) θ is periodic (with a period of 2π), (ii) the curve is defined by x(s) coordinates (not θ(s)), and (iii) there is a start-point condition but also an end-point condition for the path. These differences from the Lagrangian free particle lead to a quantum elastic equation and method to obtain a solution that are quite different from the free particle Schrödinger equation.

### 1.4. A Brief History of Elastica

The report in [18] traced the history of elastica from its first formulation by Bernoulli in 1691 to its publication in 2008. Supplementary historical notes are presented in the paper [19]. The complete solution, elastica, was usually attributed to Euler in 1744 because of his compelling mathematical treatment and illustrations, through his development of the variational method and elliptic integral theory. But, James Bernoulli came up with the correct equation a half-century earlier. Although the equation for the general elastica was published in 1695, the curves were not accurately drawn until Max Born’s 1906 doctoral dissertation [20]. Perhaps this was a sign that the elastica and the quantum method had to meet.

Mumford [21] introduced a Bayesian approach to study the elastica, where the curvature criterion is interpreted as a Brownian motion on θ(s). He derived a forward equation for the probability distribution Pr(x,θ,t) such that a curve starting at t=0 with start conditions $\left\{x^{0},\theta^{0}\right\}$ reaches {x,θ} at time *t*. He also derived a backward equation. The approach transforms the geometric deterministic problem of curve analysis into a stochastic process, allowing for probabilistic inference about curve behavior and potential curve completion in images. Mumford and others [21,22,23,24,25,26,27] used this approach to solve various computer vision problems.

### 1.5. Organization

Section 2 discusses the quantum method and its connections to some new concepts of quantum physics, namely the introduction of the Planck constant and the interpretation of measurement. It also briefly reviews the history of the path integral, which we apply as a mechanism to transform the classical elastica model into the quantum elastica model. Section 3 develops the path integral method for elastica, arriving at the quantum elastica equation. The estimation of the new parameter introduced by the quantum method is addressed. Section 4 develops the backward elastica equation and methods to obtain its solution from the forward solution. Section 5 combines the forward and backward solutions into one unique wave solution to the quantum elastica equation that simultaneously satisfies both the start boundary condition and the end boundary condition. Section 6 concludes the paper.

For readability, we put the SE(2) Fourier transform of the elastica equation in Appendix A and we present proofs of the lemmata in Appendix B.

## 2. Quantum Method Considerations Beyond Physics

Quantum physics brought to physics new properties that are not present in classical physics. We focus on the following three properties:Planck constant *h*, or the reduced one $\hbar =\frac{h}{2\pi }$.A discrete set of energy states for atoms.The uncertainty principle, where measurements of one variable, say the position of a particle, imply complete unknown values to conjugate variables, the momentum of such a particle, and lead to many discussions on the role of measurement in physics.

In order to present the quantum method as a statistical method, we address how the reduced Planck constant *ℏ* is introduced in the path integral method since it is one way to convert classical mechanics to quantum mechanics. The possible quantization of the spectrum of the Hamiltonian is discussed as a result of the statistical method. Finally, the measurement process, a subject that plays an important role throughout the development of quantum physics, is viewed as the acquisition of knowledge of a statistical method.

### 2.1. Probability Methods from Dynamical Classical Models

Consider a deterministic dynamic model M(S(t)), where $S\left(t\right)=\{s\left(t^{\prime }\right),t^{\prime }\in \left[0,t\right]\}$ describes a possible dynamic configuration path through a time interval $t^{\prime }\in \left[0,t\right]$. Given the initial boundary conditions $S\left(0\right)=s_{0}$, $S^{*}\left(t\right)=\left\{s^{*}\left(t^{\prime }\right),t^{\prime }\in \left[0,t\right]\right\}$ is the optimal configuration path, that is, $S^{*}\left(t\right)=\arg\min_{S\left(t\right)\wedge S\left(0\right)=s_{0}}M\left(S\left(t\right)\right)$, where M has units, for example, units of energy or of angular momentum or other ones. Each candidate solution path S(t) is associated with a distance to the optimal solution $D\left(S\left(t\right)\right)=M\left(S\left(t\right)\right)-M\left(S^{*}\left(t\right)\right)\geq 0$ that has the same units as M.

In order to derive a classical probabilistic model from this deterministic model, a mapping from D(S(t)) to a probability Pr(S(t))∈[0,1] is made so that the smaller D(S(t)) the larger Pr(S(t)).

An alternative non-classical probabilistic method starts with a modular distance obtained from D(S(t)), that is, the residue of D(S(t)) with respect to some modular parameter *p*, represented by $D_{p}\left(S\left(t\right)\right)=\;\mathrm{mod}\;\left(D\left(S\left(t\right)\right),p\right)$. Note that $D\left(S\left(t\right)\right)=D_{p>D(S(t))}\left(S\left(t\right)\right)$. In this case, a mapping from $D_{p}\left(S\left(t\right)\right)$ to a probability Pr(S(t))∈[0,1] is made so that the smaller $D_{p}\left(S\left(t\right)\right)$ the larger Pr(S(t)), and the concept of periodicity *p* is introduced. The path integral method, presented in the next session, has "the idea" of modular distances, but more complexity to derive the probability is introduced, as we discuss in the next section.

The purpose of presenting these probabilistic methods derived from deterministic dynamic model M(S(t)), where $S\left(t\right)=\{s\left(t^{\prime }\right),t^{\prime }\in \left[0,t\right]\}$ is that for all of them it is required that Pr(S(t)) is invariant under changes in units associated with D(). Then, a constant $k_{u}$, with the same units of D(), is introduced into the probabilistic model so that the mapping of $\frac{D_{p}\left(S\left(t\right)\right)}{k_{u}}\mapsto Pr\left(S\left(t\right)\right)$ is invariant to the choice of units to measure D().

Note that the mapping $\frac{D(S(t))}{k_{u}}\mapsto Pr\left(S\left(t\right)\right)$ is monotone for $k_{u}>0$. However, different values of $k_{u}$ lead to different probabilities.

Thus, as we argue for a quantum method to be a statistical method, the introduction and value of a statistical parameter $k_{u}$ should be addressed.

### 2.2. The Path Integral Method

Feynman developed the path integral method [13] after a suggestion by Dirac [28] to use the existing method that transforms geometric optics (classical optics in which light “chooses” to traverse a path P, from A to B, with minimum amount of time) into wave optics by assigning to each path P from A to B a phase $e^{i\frac{t(P)}{k_{t}}}$, where t=t(P) is the time that light traverses the path P, $k_{t}$ is a constant in time units, derived from the constant speed of light, so the phase is unit-less. Finally, the method sums up all possible paths that reach a coordinate X∈P in a time length *t* to construct the wave at (X,t). Thus, the Feynman-Dirac path integral method starts from the idea that the action of classical mechanics, S(P), which is the minimization criterion for selecting the optimal path of a classical particle, becomes a phase $\frac{S(P)}{\hbar }$ associated with a path. The reduced Planck constant *ℏ*, with the same units as *S*, is the QM parameter $k_{QM}=\hbar$ discussed in Section 2.1 while 2π is the modular parameter p=2π. However, the path integral method of generating the probability adds the phase term $e^{i\frac{S(P)}{\hbar }}$ for all paths to create the probability amplitude, and then by Born’s rule the magnitude square of the amplitude yields the probabilities. In order to compute the infinite sum of all the paths to obtain the probability amplitudes, Feynman proposed cutting the space of all the paths into infinitesimal paths of length δs and sum over all possible infinitesimal paths, giving an infinitesimal propagation of the wave function $\Psi_{s}\left(X\right)$ to $\Psi_{s+\delta s}\left(X\right)$, and applying this method successively to realize the full propagation. In fact, the original Dirac proposal was that $\langle \Psi_{s+\delta s}\left(X\right)|\Psi_{s}\left(X\right)\rangle =e^{i\frac{\delta S}{\hbar }}=e^{i\frac{L\delta s}{\hbar }}$, where $L(X,\dot{X},t)$ is the Lagrangian. Kac and Wiener further clarified and formalized the method [29].

We refer to this procedure as the quantum method, i.e., constructing the wave evolution via the summation of phase terms, determined by the classical action, each associated with a path from a starting condition (*A*) to an arbitrary end condition (*B*), followed by Born’s rule to derive the probability distribution. Quantum elastica refers to the quantum path integral method applied to the elastica criteria (Equation 1).

In Section 3 the development of the path integral method to the elastica problem with the start boundary condition is carried out in detail.

A novel challenge for the quantum method is posed by the elastica, namely, to combine both boundary conditions, at the start and at the end of the curve. We then consider the “anti-elastica" solution also called the backward solution, satisfying the complex conjugate equation derived for the elastica and that starts with the end condition. We combine the two solutions, the forward one that has the start condition at s=0 and the backward one that starts at the end condition and moves backward “to meet the forward one" that allows us to construct the complete wave solution. To the best of our knowledge, such a construction has not (yet) been used in physics. In Section 4 and Section 5 the inclusion of the end boundary condition in the path integral method is presented.

### 2.3. Measurements, Knowledge Acquisition, and Interpretation in the Quantum Method

The concept of measurement stands as one of the most fascinating and widely debated topics in quantum physics. In our exploration, we adhere to the interpretation which posits that particles exist in a probabilistic state and the act of measurement causes the wave function to collapse into a definite state. The Copenhagen interpretation, pioneered by Niels Bohr and Werner Heisenberg, does say that. A measurement of a quantum state yields an eigenvalue of the operator considered for the measurement, and the collapsed state is the eigenstate associated with the measured eigenvalue. The Heisenberg uncertainty principle asserts that even when a state collapses, there is still incomplete knowledge of all the variables of the state.

It is notable that the Hamiltonian derived in quantum mechanics for the Coulomb potential, yields a spectrum of discrete eignevalues for negative energies (when the electron is bounded inside the atom). This discrete set of values is the origin to the quantization of the energy levels. If we span the quantum method to other domains, quantization will occur as long as the spectrum of the Hamiltonian or other Hermitian operators (observables) admits a discrete sector.

Moreover, in the context of statistical theory, we view measurement as an acquisition of knowledge about the state of the system. This knowledge acquired from a measurement is not complete in the sense that one does not know precisely the value of all the variables associated with the collapsed state, as the uncertainty principle would now allow. One way to quantify (lack of) knowledge is via the entropy associated with the state [30]. The entropy of a quantum state must be measured in phase space where variables associated with non-commuting operators are considered (see [31]).

As a case study, the EPR experiment [32]; let us examine a version of two electrons, with opposite spins along a *z*-axis, sent in opposite directions (total momentum zero). Observer Alice measuring the spin of one electron along the *z*-axis gains knowledge that allows Alice to immediately infer the spin of the other electron along the *z*-axis. The inference is instantaneous to Alice who acquired knowledge through one measurement, but does not imply that there is any physical transmission that caused the other particle spin to collapse (after all, the speed of physical transmission is limited by the speed of light). From an information point of view, while Alice acquired knowledge of the system, another observer, Bob, that is ready to measure the other electron will not have such knowledge, and Bob predictions and measurements will reflect that. In summary, different observers will make different predictions for an experiment according to the knowledge they acquired during the experiment, and both will be right, see [33]. The quantum method, a statistical and information method, is consistent with measurements.

For elastica, the uncertainty variable associated with θ, the conjugate of θ, is(2)$$\begin{array}{c} 2\;k\left(s\right)=\frac{\partial L}{\partial \dot{\theta }}=2\dot{\theta }\left(s\right) \end{array}$$
which leads to a curvature quantum operator(3)$$\begin{array}{c} \hat{k}=-i\frac{\hbar_{e}}{2}\frac{\partial }{\partial \theta } \end{array}$$
Thus, the quantum method says that it is not possible to acquire the full knowledge for both quantities θ(s),k(s) at the same arc length *s*. Is this really true for some domain where the elastica is applied ? When using the quantum method to model the statistics of the problem, the answer is yes. One must investigate the domain of application and the uncertainties to consider a statistical method. So, the challenge of gaining complete knowledge of one or both variables can only be discussed with the domain of application and the rational behind the need for a statistical method. But in principle, why would it not be possible for a domain uncertainty to be such that one cannot know both variables for the same arc length *s*? Or in another way to present this dilemma, considering the advantages of the quantum method versus the classical statistical method, one may have to pay the price of not being able to know both variables at the same *s*. This may not be a bad price to pay if the advantages of the method outperform the use of the classical statistical method. Note that the variables *x* and θ, which will describe the coordinates of the elastica, do not form the uncertainty relation. The uncertainty associated to *x* will become clear through the Fourier transform of the quantum Hamiltonian derived in Appendix A. It is also noticeable that the elastica Hamiltonian (see Appendix A) admits a discrete spectrum on the sector of the Fourier variables n,m, and so “quantization” will also occur.

Further study of this topic for the elastica is beyond the scope of the present paper, which aims to show that the quantum method leads to an equation for the elastica quantum wave, similarly to the arrival of the Schröedinger equation from classical deterministic mechanics and quantization also occurs. We also discuss some properties of the solution, given that we have boundary conditions at the start and at the end of the elastica curve. However, in this paper, we neither attempt to analyze the solution to the equation nor apply it to any domain where evaluation of the probabilities would occur. Nevertheless, such aspects of the quantum method are worth pointing out, and an empirical study comparing knowledge acquisition by the quantum method versus knowledge acquisition in the classical statistical method is left for future work.

## 3. Quantum Elastica Equation (QEE)

In this section, we apply the path integral method to the elastica. We closely follow the technique Feynman used for reaching Schrödinger equation from classical mechanics [13]. We will first discretize the elastica action (Equation 1) in four steps, using the notation that the subindex *i* is an integer and not the imaginary number, as follows:1.We discretize the path (arc length) into *n* equal-length intervals $\epsilon =\frac{L}{n}=s_{i}-s_{i-1}$, with $s_{0}=0,s_{n}=L$.2.In the limit, n→∞; i.e., ϵ→0. Then, $\theta \left(s_{i-1}\right)=\theta \left(s_{i}-\epsilon \right)\approx \theta \left(s_{i}\right)-\epsilon \kappa_{i}$, and $x\left(s_{i-1}\right)=x\left(s_{i}-\epsilon \right)\approx x\left(s_{i}\right)-\epsilon \;e_{\theta (s_{i})}$, where $e_{\theta (s_{i})}=\left(\cos\theta \left(s_{i}\right),\sin\theta \left(s_{i}\right)\right)$, i.e,(4)$$\left[\begin{array}{c} x\left(s_{i}\right)-x\left(s_{i-1}\right)\\ \theta \left(s_{i}\right)-\theta \left(s_{i-1}\right) \end{array}\right]\approx \epsilon \;\left[\begin{array}{c} e_{\theta (s_{i})}\\ \kappa (s_{i}) \end{array}\right].$$3.Given the elastica action (Equation 1), the infinitesimal action element from $s_{i-1}$ to $s_{i}$ is(5)$$\begin{array}{c} dS_{i}\left(\epsilon \right)=\;{\left[\frac{\theta_{i}-\theta_{i-1}}{\epsilon }\right]}^{2}\epsilon =\;\frac{{\left(\theta_{i}-\theta_{i-1}\right)}^{2}}{\epsilon } \end{array}$$
where $x_{i}=x\left(s_{i}\right)$ and $\theta_{i}=\theta \left(s_{i}\right)$.4.So, the elastica action can be written as(6)$$S\left(\Gamma \right)=\lim_{\begin{array}{c} n\to \infty \\ n\;\epsilon =L \end{array}}\sum_{i=1}^{n}dS_{i}=\lim_{\begin{array}{c} n\to \infty \\ n\;\epsilon =L \end{array}}\sum_{i=1}^{n}\left[\;\frac{{\left(\theta_{i}-\theta_{i-1}\right)}^{2}}{\epsilon }\right].$$
Following the path integral method, we create the infinitesimal propagator(7)$$\begin{array}{cc} U_{\epsilon }\left(x_{i},\theta_{i};s_{i}:x_{i-1},\theta_{i-1};s_{i-1}\right) & =\lim_{\epsilon \to 0}A\left(\epsilon \right)\;e^{i\;\frac{dS_{i}\left(\epsilon \right)}{\hbar_{e}}\;} \end{array}$$
with normalization$$\begin{array}{cc} 1 & =\lim_{\epsilon \to 0}\int_{-\infty }^{\infty }A\left(\epsilon \right)\;e^{-\;\frac{\epsilon }{i\;\hbar_{e}}\;\kappa_{i}^{2}}\;d\kappa_{i}\;\Rightarrow \;A\left(\epsilon \right)=\sqrt{\frac{\epsilon }{2\;\pi \;\hbar_{e}}}\;\left(1-i\right) \end{array}$$
which evolves a wave function $\Psi_{s_{i-1}}\left(x_{i-1},\theta_{i-1}\right)$ to $\Psi_{s_{i}}\left(x_{i},\theta_{i}\right)$. The wave $\Psi_{s_{i}}\left(x_{i},\theta_{i}\right)$ is a function of the Hilbert space in the variables $\left(x,\theta \right)\in R^{2}\times S^{1}$. Note that the Hilbert space in $\theta \in S^{1}$ requires the periodic functions to be square-integrable, and the inner product of two periodic functions is the integral of their product. We rewrite the wave function at i−1 using (Equation 4)(8)$$\begin{array}{c} \Psi_{s_{i-1}}\left(x_{i-1},\theta_{i-1},\right)\approx \Psi_{s_{i}-\epsilon }\left(x_{i}-\epsilon \;e_{\theta_{i}},\theta_{i}-\epsilon \;\kappa_{i}\right) \end{array}$$
and the infinitesimal propagation of the wave function as(9)$$\begin{array}{cc} \Psi_{s_{i}}\left(x_{i},\theta_{i}\right) & =\lim_{\epsilon \to 0}\int_{-\infty }^{\infty }\;A\left(\epsilon \right)\;e^{i\;\frac{1}{\hbar_{e}}\;\frac{{\left(\theta_{i}-\theta_{i-1}\right)}^{2}}{\epsilon }}\;\Psi_{s_{i-1}}\left(x_{i-1},\theta_{i-1}\right)\;\frac{1}{\epsilon }d\theta_{i-1}\\ \; & =\lim_{\epsilon \to 0}\int_{-\infty }^{\infty }\;A\left(\epsilon \right)\;e^{i\;\frac{\epsilon }{\hbar_{e}}\;\kappa_{i}^{2}}\;\Psi_{s_{i}-\epsilon }\left(x_{i}-\epsilon \;e_{\theta_{i}},\theta_{i}-\epsilon \;\kappa_{i}\right)\;d\kappa_{i} \end{array}$$

To compute the integral in Equation (Equation 9), we expand the wave function $\Psi (x_{i}-\epsilon \;e_{\theta_{i}},\theta_{i}-\epsilon \;\kappa_{i},s_{i}-\epsilon )$ around (x,θ,s−ϵ), dropping the index “*i*” and letting $\hat{e}=e_{\theta }\left(s\right)\cdot \nabla_{x}$.(10)$$\begin{array}{c} \Psi_{s-\epsilon }\left(x-\epsilon \;e_{\theta }\left(s\right),\theta -\epsilon \;\kappa \right)\approx \;\\ \;[1-\epsilon \;\hat{e}+\frac{1}{2!}\epsilon^{2}\;{\hat{e}}^{2}-\frac{1}{3!}\epsilon^{3}\;{\hat{e}}^{3}+\ldots +\epsilon^{2}\;\kappa \;\hat{e}\;\frac{\partial }{\partial \theta }-\frac{1}{2!}\epsilon^{3}\;\left(\kappa^{2}\hat{e}\;\frac{\partial^{2}}{\partial \theta^{2}}+\kappa \;{\hat{e}}^{2}\;\frac{\partial }{\partial \theta }\right)-\ldots \\ -\epsilon \;\kappa \frac{\partial }{\partial \theta }+\frac{1}{2!}{\left(\epsilon \;\kappa \right)}^{2}\frac{\partial^{2}}{\partial \theta^{2}}-\frac{1}{3!}{\left(\epsilon \;\kappa \right)}^{3}\frac{\partial^{3}}{\partial \theta^{3}}+\ldots ]\;\Psi_{s-\epsilon }\left(x,\theta \right) \end{array}$$

Plugging this Taylor expansion into Equation (Equation 9) leads to a series of integrals on the curvature κ for each term. All odd terms on κ vanish since the integrand is an odd function, i.e., $0=\int_{-\infty }^{\infty }e^{-\;\frac{\epsilon }{i\;\hbar_{e}}\;\kappa^{2}}\;\kappa^{2p+1}\;d\kappa$, with *p* being an integer. The terms in zero order on κ have the integral to yield $1=A\left(\epsilon \right)\;\int_{-\infty }^{\infty }e^{i\;\alpha \;\kappa^{2}}\;d\kappa$, so, up to the first order of ϵ, we obtain the contribution from the terms $\left(1-\epsilon \;(e_{\theta }\left(s\right)\cdot \nabla_{x})\right)$. The first quadratic term in κ provides (defining $\alpha =\frac{\epsilon }{\hbar_{e}}>0$)(11)$$\begin{array}{cc} A\left(\epsilon \right)\;\int_{-\infty }^{\infty }e^{i\;\alpha \;\kappa^{2}}\;\frac{1}{2!}\;\epsilon^{2}\kappa^{2}\;d\kappa \;\frac{\partial^{2}}{\partial \theta^{2}}\; & =\frac{\epsilon^{2}}{2!}\;A_{\epsilon }\;\left[\;\frac{\partial }{\partial (i\alpha )}\int_{-\infty }^{\infty }e^{i\;\alpha \;\kappa^{2}}\;d\kappa \right]\;\frac{\partial^{2}}{\partial \theta^{2}}\\ \; & =\frac{\epsilon^{2}}{2!}\;A_{\epsilon }\;\left(i-1\right)\sqrt{\frac{\pi }{8\;\alpha^{3}}}\;\frac{\partial^{2}}{\partial \theta^{2}}\\ \; & \;\mathrm{back}\;\mathrm{to}\;\;\alpha =\frac{\epsilon }{\hbar_{e}}\;\mathrm{and}\;A\left(\epsilon \right)\;=\sqrt{\frac{\epsilon }{2\;\pi \;\hbar_{e}}}\;\left(1-i\right)\\ \; & =\frac{\epsilon }{4}\;\left(-i\hbar_{e}\right)\;\;\frac{\partial^{2}}{\partial \theta^{2}} \end{array}$$
which is linear in ϵ. All other quadratic terms in κ or terms with even and higher powers of κ will have larger powers of ϵ dependencies. We then write Equation (Equation 9), keeping the terms up to first order in ϵ to obtain(12)$$\begin{array}{cc} \Psi_{s}\left(x,\theta \right) & \approx \left[\;1-\epsilon \;e_{\theta }\cdot \nabla_{x}-\frac{1}{4}\;i\;\epsilon \;\hbar_{e}\;\frac{\partial^{2}}{\partial \theta^{2}}\;\right]\;\Psi_{s-\epsilon }\left(x,\theta \right)\\ \; & ⇓\\ \Psi_{s}\left(x,\theta \right)-\Psi_{s-\epsilon }\left(x,\theta \right) & \approx \left[\;-\;\epsilon \;e_{\theta }\cdot \nabla_{x}-\frac{1}{4}\;i\;\epsilon \;\hbar_{e}\;\frac{\partial^{2}}{\partial \theta^{2}}\;\right]\;\Psi_{s-\epsilon }\left(x,\theta \right) \end{array}$$
Multiplying Equation (Equation 12) by $i\;\hbar_{e}$, dividing by ϵ, rearranging, and then taking the limit ϵ→0, we obtain the quantum elastica equation(13)$$\begin{array}{cc} i\;\hbar_{e}\;\frac{\partial }{\partial \;s}\Psi_{s}\left(x,\theta \right) & =H\;\Psi_{s}\left(x,\theta \right) \end{array}$$
where we interpreted *s* as a time parameter to define a Hamiltonian as(14)$$\begin{array}{c} H=\left[-\;i\;\hbar_{e}\;e_{\theta }\cdot \nabla_{x}+\;\frac{\hbar_{e}^{2}}{4}\frac{\partial^{2}}{\partial \theta^{2}}\;\right]\;. \end{array}$$
Note that, from (Equation 3), the second term is the square of the curvature operator (the “kinectic” energy of the elastica). The first term reflects the complexity of the elastica model, where the conjugate of the position operator $-\;i\;\hbar_{e}\;\nabla_{x}$ interacts with the angle of the curve. Notation-wise, we move the parametrization index *s* to $\Psi \left(x,\theta ,s\right)=\Psi_{s}\left(x,\theta \right)$.

### 3.1. The Parameter $\hbar_{e}$ and Wick Rotation

QM introduces a parameter *ℏ* not present in the classical mechanics modeling, with the same units as the action, energy–time. In quantum mechanics, the quantum of action is the constant $\hbar =1.054571817\times 10^{-34}$ joule seconds, obtained by Planck studying black body radiation and noticing that the energy of each photon was quantized in packages of energy E=ℏω, where ω=2πf is the frequency of a given photon.

In statistical mechanics, a constant introduced into probability models is $k_{B}$—the Boltzmann constant. The Boltzmann constant connects the kinetic energy to the thermal temperature, where the average kinetic energy of a particle in a gas per degree of freedom is $\frac{1}{2}k_{B}T$, and *T* is the kelvin temperature, and then $k_{B}=1.380649\times 10^{-23}$ joules per kelvin.

Moreover, Wick rotation [34] “converts” quantum physics to statistical mechanics by associating the unitary evolution operator $e^{-i\frac{H}{\hbar_{e}}t}$ with the Boltzmann distribution $e^{-\beta H}$ by substitution $\frac{1}{k_{B}T}\to i\frac{t}{\hbar_{e}}$. The Wick rotation provides a connection between thermodynamics and quantum field theory, e.g., see [35,36]. Thus, we stress the relation of these constants which were introduced in order to derive these statistical models from the same classical deterministic model.

Similarly to the relation between quantum mechanics and classical statistical mechanics (thermodynamics) we can derive Mumford’s classical statistical elastica equation [21] from the QEE (Equation 13) via a Wick-like rotation that maps $i\frac{1}{\hbar_{e}}\mapsto \frac{1}{4\sigma }$ to yield$$\begin{array}{cc} \frac{\partial }{\partial \;s}Pr_{s}\left(x,\theta \right) & =\left[-\;\;e_{\theta }\cdot \nabla_{x}+\;\sigma \frac{\partial^{2}}{\partial \theta^{2}}\;\right]\;Pr_{s}\left(x,\theta \right)\;, \end{array}$$
The parameter σ in the statistical elastica formulation represents the standard deviation of the curvature (the standard deviation of the Brownian motion). This value depends on the domain of application of the elastica model, just as $\hbar_{e}$ depends on the domain of application.

For the elastica problem, similar to quantum mechanics, the quantum Hamiltonian also produces a spectrum of eigenvalues that are scaled by the parameter $\hbar_{e}$, which is more clearly seen in the Fourier version of the Hamiltonian (Equation 27). The initial condition defines the expected value of the energy (which must be the same on both boundary conditions), and the unitary evolution of the elastica guarantees energy conservation. The challenge of obtaining a real value for $\hbar_{e}$ is to obtain solutions to the elastica and estimate the expected energy for it, which will depend on $\hbar_{e}$, and compare it with empirical measurements. This value will vary according to the domain of application of the elastica model. For the mathematical elastica problem, without a domain of application, any value of $\hbar_{e}>0$ provides a solution to the elastica.

So, in order for the quantum method to assign a value $\hbar_{e}$ to a classical problem, one needs to identify the domain of application of the model, extract the expected value of the Hamiltonian at the initial condition, and compare it to empirical values of such energy in such a domain.

### 3.2. Some Properties of the QEE

The elastica are equivariant under the SE(2) group, transformations of space that preserve Euclidean distance, not including reflections, i.e., compositions of rotation and translation. This is straightforward to verify since rotations and translations do not change length and curvature.

**Lemma 1.** *The Hamiltonian* (Equation 14) *is Hermitian*.

This is a key property of a Hamiltonian in quantum physics, guaranteeing that all eigenvalues are real values and that the unitary wave function evolution is the same as the time-reversed one.

**Definition 1** (Parity Transformation (P))**.**
(15)$$\begin{array}{c} x\to x^{\prime }=T-x\;\mathrm{and}\;\theta \to \theta^{\prime }=\mathrm{mod}\left(\theta +\pi ,2\pi \right)\;. \end{array}$$
*where T is any spatial translation.*

**Lemma 2.** *The Hamiltonian* (Equation 14) *is invariant under the parity transformation (P). Moreover, if Ψ(x,θ,s) is a solution to* (Equation 13), *then the wave function*
$$\Psi^{P}\left(x,\theta ,s\right)=P\left(\Psi \left(x,\theta ,s\right)\right)=\Psi \left(T-x,\mathrm{mod}\left(\theta +\pi ,2\pi \right),s\right)$$
*also satisfies* (Equation 13).

Figure 1 provides a visualization of a parity transformation coordinate system with more detailed descriptions of some of its properties.

**Lemma 3.** 
*The Hamiltonian is dispersive. More precisely, it is dispersive in θ, and spatial diffusion occurs through the interaction between θ and the spatial coordinate angle ϕ, where x=r(cosϕ,sinϕ).*


As discussed earlier, measurements in the quantum method can be viewed as knowledge acquisition, and (lack of) knowledge is quantified by the entropy in phase space. We then define the entropy for the elastica.

**Definition 2** (Entropy in Phase Space)**.** $$S\left(s\right)=-\int_{0}^{2\pi }d\theta \;\;\int \int \;d^{2}{x\;|\Psi \left(x,\theta ,s\right)|}^{2}{\ln|\Psi \left(x,\theta ,s\right)|}^{2}+\sum_{m,n}\int_{0}^{\infty }|{\tilde{\Psi }}_{n,m}{\left(k,s\right)|}^{2}\ln{\left|{\tilde{\Psi }}_{n,m}\left(k,s\right)\right|}^{2}\;k\;dk\;,$$
*where ${\tilde{\Psi }}_{n,m}\left(k,s\right)$ is the SE(2) Fourier transform of Ψ(x,θ,s), see* (Equation 24). *The entropy is defined in the phase space associated with the coordinate space $\left(x,\theta \right)\in R^{2}\times \left[0,2\pi \right)$ in analogy to the phase space entropy in quantum physics [31]. The Fourier to the periodic variable θ is the integer variable n, while the Fourier to x=r(cosϕ,sinϕ) are the variables (m,k). If one measures x, the values of (m,k) become unknown and vice versa.*

**Conjecture 1.** *We conjecture that, for the elastica solution Ψ(x,θ,s) with start boundary condition* (Equation 16), *and as s increases, the entropy S(s) defined above increases. Here, increase is in the weak sense; that is, the entropy never decreases.*

The motivation for the conjecture is, given that the start boundary condition is concentrated at one position and at one angle, Lemma 3 says that the Hamiltonian will disperse the wave; thus, we conjecture, causing the entropy to increase.

### 3.3. Boundary Conditions

In order to solve (Equation 13), a first-order derivative in *s* requires a start boundary condition. For notation purposes, we refer to the start boundary condition wave as(16)$$\begin{array}{cc} \Psi \left(x,\theta ,0;x^{0},\theta^{0}\right)=\Psi_{s=0}\left(x,\theta ;x^{0},\theta^{0}\right) & =\delta \left(x-x^{0}\right)\delta \left(\theta -\theta^{0}\right)\; \end{array}$$
where the Dirac δ(.) function is to be interpreted as a function in Hilbert space that is sharply concentrated in the start conditions. This abuse of notation will have no impact on the formalism developed.

The elastica solution also requires an end condition in $\left(x^{L},\theta^{L}\right)$; i.e., at s=L, the solution is concentrated in $\left(x^{L},\theta^{L}\right)$. Note that there are paths that start at $\left(x^{0},\theta^{0}\right)$ and travel for a length *s*, contributing to the wave function Ψ(x,θ,s), but are further than L−s away from the end condition ($|x-x^{L}|\;>\;L-s$) and therefore should not end up contributing to the final wave function (see Figure 2a).

The final solution we are looking for, $\Psi^{L}\left(x,\theta ;x^{0},\theta^{0},x^{L},\theta^{L}\right)$, is associated with curves of length *L* that started and ended at the boundary conditions so that $P^{L}\left(x,\theta \right)=\left|\Psi^{L}\left(x,\theta ;x^{0},\theta^{0},x^{L},\theta^{L}\right)\right|$ can be interpreted as the probability of an elastica curve, of length *L* and boundary conditions $\left(x^{0},\theta^{0},x^{L},\theta^{L}\right)$, to go through (x,θ). This solution may be a superposition of waves $\phi_{s}^{L}\left(x,\theta ,s;x^{0},\theta^{0},x^{L},\theta^{L}\right)$ that evolve for a total length *L* and, while it may disperse for some length *s*, it must change the dispersion behavior; that is, it must evolve towards the end of the length *L* concentrating on the end condition $\left(x^{L},\theta^{L}\right)$. In the next section, we will develop a necessary step in constructing such a solution, which will finally be achieved in Section 5.

## 4. The Backward QEE

Let $T=x^{L}+x^{0}$ and$$\Psi^{P}\left(x,\theta ,s\right)=P\left(\Psi \left(x,\theta ,s\right)\right)=\Psi \left(x^{L}+x^{0}-x,\mathrm{mod}\left(\theta +\pi ,2\pi \right),s\right)\;.$$
See Figure 1 and Figure 2a. As shown in Lemma 2, the wave $\Psi^{P}\left(x,\theta ,s\right)$ satisfies (Equation 13). Following quantum theory, we apply the conjugate transpose operation to (Equation 13); i.e.,(17)$$\begin{array}{cc} -i\;\hbar_{e}\;\frac{\partial }{\partial \;s}\left(\Psi^{P,*}\right)\left(x,\theta ,s\right) & =H^{†}\left(\Psi^{P,*}\right)\left(x,\theta ,s\right)\\ \; & ⇓H\;\mathrm{is}\mathrm{Hermitian}\\ -i\;\hbar_{e}\;\frac{\partial }{\partial \;s}\Psi^{\mathrm{CP}}\left(x,\theta ,s\right) & =H\;\Psi^{\mathrm{CP}}\left(x,\theta ,s\right) \end{array}$$
where we defined $\Psi^{\mathrm{CP}}\left(x,\theta ,s\right)=\Psi^{P,*}\left(x,\theta ,s\right)$ to be the complex conjugate of $\Psi^{P}$. We refer to (Equation 17) as the backward equation.

**Lemma 4.** *Assume Ψ(x,θ,s) satisfies the quantum elastic Equation* (Equation 13) *and is described by a coordinate system transformed by P with $T=x^{L}+x^{0}$ and starting boundary condition $\left(x^{0},\mathrm{mod}\left(\theta^{L}+\pi ,2\pi \right)\right)$ (see Figure 1). Then, $\Psi^{\mathrm{CP}}\left(x,\theta ,s\right)$ satisfies* (Equation 17) *with the starting boundary condition $\left(x^{L},\theta^{L}\right)$.*

Define the “length reversal” transformation (T) to be $s^{\prime }=T\left(s\right)=L-s$. Then, define(18)$$\begin{array}{c} \Psi^{\mathrm{CPT}}\left(x,\theta ,s\right)=T\left(\Psi^{\mathrm{CP}}\left(x,\theta ,s\right)\right)=\Psi^{\mathrm{CP}}\left(x,\theta ,L-s\right)\;. \end{array}$$

**Lemma 5.** *Assume $\Psi^{\mathrm{CP}}\left(x,\theta ,s\right)$ satisfies the backward Equation* (Equation 17), *and, at s=0, it satisfies the end condition $\left(x^{L},\theta^{L}\right)$. Then, $\Psi^{\mathrm{CPT}}\left(x,\theta ,s\right)$ satisfies the QEE* (Equation 13) *and at s=L it satisfies the end condition $\left(x^{L},\theta^{L}\right)$.*

We therefore have constructed a solution $\Psi^{\mathrm{CPT}}\left(x,\theta ,s\right)$ that starts at (x,θ) and arc length *s* and moves forward in *s*, focusing so that, at s=L, it is concentrated at the end condition.

Figure 2 illustrates typical path segments that are associated with the wave functions $\Psi^{\mathrm{CP}}\left(x,\theta ,s\right)$ and $\Psi^{\mathrm{CPT}}\left(x,\theta ,s\right)$.

The evolution of the conjugate solution $\Psi^{\mathrm{CP}}\left(x,\theta ,s\right)$ that moves back in *s* and the associated solution $\Psi^{\mathrm{CPT}}\left(x,\theta ,s\right)$ that moves forward in *s* appears in physics in the Feynman–Stückelberg interpretation of the antiparticle solution [37,38]. Moreover, due to (i) quantum probabilities (Born rule) being invariant to complex conjugate operations and (ii) the wave solution being a scalar (there is no “charge” associated with the wave), both waves Ψ(x,θ,s) moving forward in *s* and the conjugate $\Psi^{\mathrm{CPT}}\left(x,\theta ,s\right)$ are to be interpreted as representing the same type of “elastica”; i.e., the anti-elastica is an elastica just as an anti-photon is a photon.

We also note that, if Conjecture 1 is valid, then the solution $\Psi^{\mathrm{CP}}\left(x,\theta ,s\right)$ will have the entropy to decrease as it evolves along *s* to s=L when the solution concentrates in the boundary condition at the end.

## 5. Elastica Quantum Wave and Probabilities

In this section, we derive a quantum wave solution to elastica that evolved to length *L*, satisfying both the start–end conditions.

Consider the path integral that describes the sum over all the paths that start at the initial condition $\left(x^{0},\theta^{0}\right)$, reach (x,θ) with length *s*, and continue from (x,θ) for a length L−s, reaching the end at $\left(x^{L},\theta^{L}\right)$.(19)$$\begin{array}{c} \Phi^{L}\left(x,\theta ,s;x^{0},\theta^{0},x^{L},\theta^{L}\right)=\frac{1}{Z_{s}}\sum_{m_{s}\in M_{s}}\sum_{n_{s}\in N_{L-s}}e^{i\frac{1}{\hbar_{e}}\left(S_{m_{s}}\left(x,\theta ;x^{0},\theta^{0}\right)+S_{n_{s}}\left(x,\theta ;x^{L},\theta^{L}\right)\right)} \end{array}$$
where $M_{s}$ is the set of segments of Γs with length *s*; i.e., for all $m_{s}\in M_{s}$, $m_{s}\left(s=0\right)=\left(x^{0},\theta^{0}\right)$ and $m_{s}\left(s\right)=\left(x,\theta \right)$, while $N_{L-s}$ is the set of segments of Γs with length L−s that, for all $n_{s}\in N_{L-s}$, $n_{s}\left(s=0\right)=\left(x,\theta \right)$ and $n_{s}\left(L-s\right)=\left(x^{L},\theta^{L}\right)$, and $S_{P}$ is the action of each segment P. Any such path of length *L* is a concatenation of segments $m_{s}$ and $n_{s}$; i.e., $\left\{m_{s}\left(s^{\prime }\right)=\Gamma \left(s^{\prime }\right);s^{\prime }=\left[0,s\right]\right\}$ and $\{n_{s}\left(s^{\prime }-s\right)=\Gamma \left(s\right);s^{\prime }=\left[s,L\right]\}.$ The sets $M_{s}$ and $N_{L-s}$ are clearly disjoint for a given *s*, and their union forms the set of all paths Γ that have length *L* and both boundary conditions. Figure 2 depicts a path of length *L* with both boundary conditions and divided into two segments, one of length *s* and the other of length L−s. Finally, $Z_{s}$ is a normalization that is needed since two boundary conditions were imposed, not one. The normalization is over the entire space, $\left(x,\theta \right)\in R^{2}\times S^{1}$, and therefore will depend on the index *s*.

Thus, (Equation 19) describes a wave solution to QEE that evolved for a length *L*, having started at the boundary condition $\left(x^{0},\theta^{0}\right)$, dispersed for an arc length *s* reaching (x,θ), and then started to converge (in contrast to disperse) for an arc length L−s when it ended at the boundary condition $\left(x^{L},\theta^{L}\right)$.

A rule of thumb in classical and quantum probability is that propositions joined by a logical AND are combined through multiplication. In the case of quantum probabilities, such operations occur with quantum states and not with probabilities.

**Lemma 6.** *The quantum wave* (Equation 19) *can be obtained as*
(20)$$\begin{array}{cc} \Phi^{L}\left(x,\theta ,s;x^{0},\theta^{0},x^{L},\theta^{L}\right) & =\frac{1}{Z_{s}}\Psi \left(x,\theta ,s;x^{0},\theta^{0}\right)\;\Psi^{\mathrm{CPT}}\left(x,\theta ,s;x^{L},\theta^{L}\right)\; \end{array}$$

Normalization $Z_{s}$ is needed since each wave component, $\Psi (x,\theta ,s;x^{0},\theta^{0})$ and $\Psi^{\mathrm{CPT}}\left(x,\theta ,s;x^{L},\theta^{L}\right)$, with the respective boundary conditions, is already normalized, so the product of them is not.

Here, different from the integral path formulation for classical mechanics, the two boundary conditions imply that a given path of length *L*, say $\Gamma^{L}$, and given boundary conditions $\Gamma^{L}\left(s=0\right)=\left(x^{0},\theta^{0}\right);\Gamma^{L}\left(s=L\right)=\left(x^{L},\theta^{L}\right)$, will contribute to different final waves $\Phi^{L}\left(x,\theta ,s;x^{0},\theta^{0},x^{L},\theta^{L}\right)$, indexed by *s*, in different ways. More precisely, considering all paths of length *L* and focusing on a coordinate (x,θ): the wave produced based on the forward diffusion for an arc length *s* only absorbed the action associated with all $\Gamma^{L}$ that reached (x,θ) at length *s*, while a different wave produced based on the forward diffusion for an arc length $s^{\prime }\neq s$ absorbed the action associated with all $\Gamma^{L}$ that reached (x,θ) at length $s^{\prime }\neq s$. Thus, different wave solutions from the same set of paths with two boundary conditions are generated by the path integral method; each one is indexed by *s*. One way to illustrate the scenario is to associate each index *s* with a different color, so the set of all paths that satisfy the boundary conditions and have total length *L* are colored differently with the color index *s*, yielding different color wave functions $\phi_{s}$.

The path integral method with two boundary conditions, at the start and at the end, must then combine all these possible waves, each of which is a different solution to the problem. A rule of thumb in counting in classical probabilities and in quantum probabilities is that propositions where the classical logic is OR combine via addition. In the case of quantum probabilities, such operations occur with quantum states and not with probabilities.

We argue that the path integral method that considers all possible solutions creates a superposition of these waves to produce the final wave solution as follows:

**Proposition 1.** 
*The quantum wave*

(21)
$$\begin{array}{cc} \Psi^{L}\left(x,\theta ;x^{0},\theta^{0},x^{L},\theta^{L}\right) & =\frac{1}{Z}\int_{0}^{L}\Phi^{L}\left(x,\theta ,s;x^{0},\theta^{0},x^{L},\theta^{L}\right)\;ds\\ \; & =\frac{1}{Z}\int_{0}^{L}\frac{1}{Z_{s}}\Psi \left(x,\theta ,s;x^{0},\theta^{0}\right)\;\Psi^{\mathrm{CPT}}\left(x,\theta ,s;x^{L},\theta^{L}\right)\;ds\;. \end{array}$$

*where Z is a normalization constant, describes a wave solution to the QEE that evolved for a length L, having started concentrated at boundary condition $\left(x^{0},\theta^{0}\right)$ and ended concentrated at the boundary condition $\left(x^{L},\theta^{L}\right)$. $P^{L}\left(x,\theta \right)={\left|\Psi^{L}\left(x,\theta ;x^{0},\theta^{0},x^{L},\theta^{L}\right)\right|}^{2}$ is the quantum probability of the elastica of length L to go through (x,θ), having the start–end boundary conditions $\left\{\left(x^{0},\theta^{0}\right);\left(x^{L},\theta^{L}\right)\right\}$.*


Thus, with respect to the original elastica problem, the quantum method does not recover curves of length *L*; instead, it recovers local probabilities $P^{L}\left(x,\theta \right)$ that the elastica of length *L* with start–end boundary conditions passes through (x,θ).

It is plausible, but requires more study, that, as $\hbar_{e}\to 0$, the trace of locations (x,θ) that have non-zero probability will be the elastica curve.

## 6. Conclusions

This work extends the quantum method to optimization problems, not limited to physics, that are characterized by action minimization. To demonstrate this extension, we applied the quantum method to the elastica problem, a classic variational calculus problem with a rich mathematical history, notably explored in Max Born’s PhD thesis [20]. We posit that the elastica problem serves as a prototypical example for applying quantum methods to variational problems, providing a framework for deriving quantum algorithms (quantum equations) across a broad range of non-physical problems.

Compared to classical statistical approaches to optimization, the quantum method offers potentially enhanced robustness. This is evidenced by quantum theory’s provision of a stable and robust description of atomic structure in physics, as well as its superior performance over classical statistical methods in image processing tasks such as circle detection [15,17]. This enhanced robustness stems from the capacity of waves, and hence quantum probabilities, to exhibit destructive interference, a phenomenon that is absent in classical statistical frameworks. However, this paper does not empirically assess solution robustness; it focuses instead on the derivation of quantum equations and solutions for the elastica problem. Also, the need for the introduction of a new parameter $\hbar_{e}$, in analogy to the introduction of *ℏ* in quantum mechanics (since it is absent in classical mechanics), is a consequence of extending a deterministic model to a probability model where any solution must be invariant to the choice in units used. It is quite remarkable that the entire quantum field theory of the standard model requires just one such parameter, *ℏ*, suggesting that the entire theory should have one classical deterministic model from which quantum field theory can be derived. However, for the elastica, empirical observations of the energy of the solution are required to obtain the exact value of this new parameter. The elastica was treated as a mathematical problem and the domain of application was not chosen. What are the empirical observations of the energy of an elastica? This work is left as future work, an empirical study of the solution of the quantum elastica equation for a given domain.

Our key contribution to the quantum method is the incorporation of boundary conditions at both ends of the elastic curve by combining forward and backward wave solutions via multiplication. The forward solution incorporates the initial condition, while the backward solution (or “anti-elastica” solution) incorporates the end condition. Although the forward solution disperses the initial condition, the backward solution refocuses this dispersed wave toward the terminal condition.

Solving such equations can be computationally expensive due to interference effects. Unlike classical probabilities, where truncation is often feasible, the wave nature of interference means that the phase of every wave term, regardless of amplitude, can influence the final probability values. Consequently, the superior robustness and accuracy of the quantum method are counterbalanced by increased computational demands. Quantum computers, which leverage inherently quantum physical processes, are expected to address this computational burden significantly more efficiently [39,40,41].

## Figures and Tables

**Figure 1 entropy-27-00388-f001:**
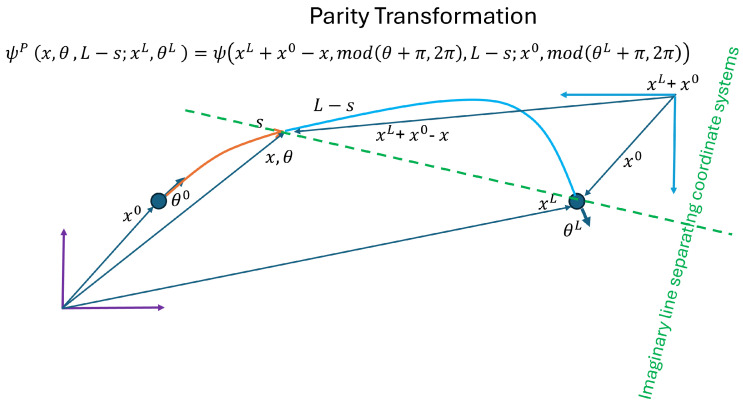
Parity transformation for $T=x^{L}+x^{0}$. The vector *x* describes a point in the purple coordinate system where the orange and cyan segments meet, while the vector $x^{L}+x^{0}-x$ describes the same point in the cyan coordinate system with origin at $x^{L}+x^{0}$ and its axes reversed. The two coordinate systems are “separated by an imaginary green dashed line”, stressing that *x* and $x^{L}+x^{0}-x$ refer to the same point written in two different coordinate systems. In this cyan reversed coordinate system, the angles are also described as if they are in opposite directions, i.e., θ→mod(θ+π,2π). Note that, while Ψ incorporates the solid blue segment described in the reversed coordinate system (in cyan), including the end point being described by $\left(x^{0},\mathrm{mod}\left(\theta^{L}+\pi ,2\pi \right)\right)$, $\Psi^{P}$ incorporates the same solid blue segment described in the original coordinate system (purple).

**Figure 2 entropy-27-00388-f002:**
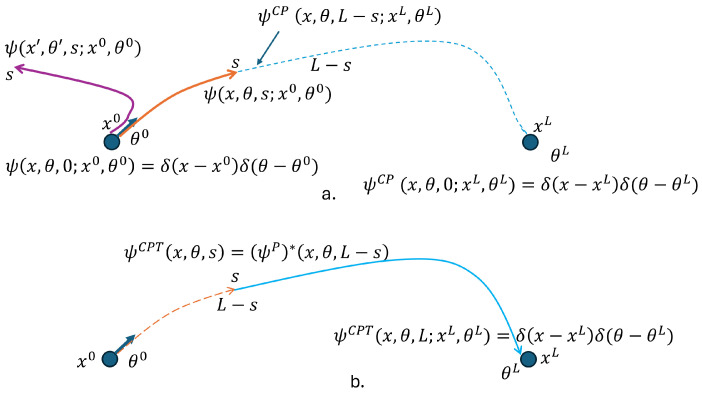
(**a**) Depicts two elastica curves with start condition at $\left(x^{0},\theta^{0}\right)$. One has length *L* and is split in two segments, where the solid orange segment with length *s* ends at x,θ. The complement blue dashed segment with length L−s starts at x,θ and ends at $\left(x^{L},\theta^{L}\right)$. The blue dotted segment starting at $\left(x^{L},\theta^{L}\right)$ and moving backwards for a length L−s contributes to the wave $\Psi^{\mathrm{CP}}\left(x,\theta ,L-s\right)=\Psi^{*}\left(x^{L}+x^{0}-x,\mathrm{mod}\left(\theta +\pi ,2\pi \right),L-s\right)$ (see Figure 1). The other one is solid purple and reaches at *s* the coordinates $\left(x^{\prime },\theta^{\prime }\right)$. However, $|x^{\prime }-x^{L}|>L-s$. Thus, there is no elastica segment of length L−s that starts at $\left(x^{L},\theta^{L}\right)$ and reaches $\left(x^{\prime },\theta^{\prime }\right)$. (**b**) Combining two segments (dotted orange and solid blue) to describe a curve of length *L* moving forward in *s* from s=0 to s=L with start and end boundary conditions is solved by revisiting the description of the dashed blue segment of (a), with length L−s, and now coloring in solid blue to indicate that a length reversal operation occurred and now it starts at (x,θ) and ends at $\left(x^{L},\theta^{L}\right)$ (see Lemma 4). Note that, for $|x-x^{L}|<L-s-\epsilon$, there are several segments of length L−s with the same start (x,θ) and same end $\left(x^{L},\theta^{L}\right)$ that contribute to yield $\Psi^{\mathrm{CPT}}\left(x,\theta ,s\right)$. The parameter ϵ>0 reflects a needed adjustment for the tangent constraints.

## Data Availability

No datasets were generated or analyzed during the current study.

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
