# Peer review of "Quantum Elastica"

_entropy, 2025, doi:10.3390/e27040388_

Round 1

Reviewer 1 Report

Comments and Suggestions for Authors

In the manuscript the authors propose to use the Feynman path integral
approach to solve a problem of classical variational calculus: the
elastica problem, which has a long history. No doubt, as the authors
write, "Quantum mathematics, even without quantum hardware can offer
novel solutions and insights to classical optimization challenges."
This is a very promising research direction, leading to ideas such as
the use of tensor networks for generic optimization, etc. The present
paper intends to contribute to this line of research.

The exact motivation and the benefits of the method are less clear
though. In the introduction, even if the long history of the problem
is mentioned, it is not pointed out why we are in the need of new
solution methods, and what could we benefit from them. In the
conclusions it is stated that the main benefit of the method is
"superior robustness and accuracy", but this is not
demonstrated. Meanwhile it is stated that "the superior robustness and
accuracy of the quantum method are counterbalanced by increased
computational demands". Although this is followed by a vague claim
that quantum computers are expected overcome this issue, the details
of this is obscure.

In the introduction of the elastica problem, the authors formulate it
in a Lagrangian way.  The path integral method is then introduced very
briefly and rather qualitatively, albeit this is the main apparatus
used in the present contribution. The authors essentially follow the
steps that can be understood from Feynman's original work (cited as
Ref. [13], albeit the citation is wrong, the correct volume of this
paper is 20 not 2), but even this is not stated.  As the quantum
information community dominantly uses Hamiltonian formulations, this
limits the accessibility of the description to those who are more
familiar with the Feynman path integral method, even though the
authors seem to address the optimization and quantum computation
community.

Section 2 then introduces the quantum elastica equations, and then
quantum mechanics enters, rather suddenly especially for those not
familiar with path integrals. But even so, why do we need the constant
hbar appear here, if once it is not supposed to be a physical problem,
just a physical method to solve it? Some explanation should have been given.

A more important problem is the introduction and implementation of the
wave functions. It is said that the wave function is in "Hilbert
space", but no details are given about this Hilbert space? The
consideration also lacks a proper description of the measurement process
which would be the basis of the probability interpretation. Without
the definition of measurement it is questionable whether the function
P_L(x,\theta) after Equation (17) obeys a probability
interpretation. Moreover, lemma 6 states that Psi^L satisfies the
quantum elastica equations, but Eq.(11) is for a function with a
variable s, so this cannot be even substituted into it.

A similar lack of interpretation appears at other points of the
manuscript, too. For instance, phase space entropy is introduced in
Definition 1, but its operational meaning is not mentioned. Meanwhile,
the argument around Eq.(19) suggests that a known stochastic treatment
can be recovered from the present formulation, which looks like a very
interesting point that would deserve more emphasis and discussion.

Looking at the calculational details, the beginning of the proof of
Lemma 6 seems to be incorrect, when carrying out the substitution in
the unnumbered equation after equation (A13), a -((hbar^2)/4)
(\partial_\theta \Psi)(\partial_\theta \Psi^CPT) summand seems to be
missing from the right hand side. It should be at least doubly checked.

Altogether the manuscript intends to give a long-term vision of a new
and original approach, yet with not yet justified benefits.  The calculations are technically involved, but the correctness of some statements is questionable, important interpretational details are missing, and the description is less accessible. Hence, I cannot, unfortunately, recommend its publication.

Author Response

First, we want to thank you very much for the review. The feedback and corrections have greatly improved the paper by requiring clarification of important topics and correcting mistakes.

The exact motivation and the benefits of the method are less clear
though. In the introduction, even if the long history of the problem
is mentioned, it is not pointed out why we are in the need of new
solution methods, and what could we benefit from them. In the
conclusions it is stated that the main benefit of the method is
"superior robustness and accuracy", but this is not
demonstrated. Meanwhile it is stated that "the superior robustness and
accuracy of the quantum method are counterbalanced by increased
computational demands". Although this is followed by a vague claim
that quantum computers are expected overcome this issue, the details
of this is obscure.

We agree with the reviewer, and in the introduction state that the paper’s main contribution is the understanding that quantum theory is a statistical technique that can be used in different domains. We also showed that through Wick rotations new quantum method  map to classical statistics method. Wick rotation is a well know tool in quantum physics to link to statistical (classical) physics and so Wick rotation is part of the tools of quantum mathematics.   We have “tuned down” the narrative about the benefits of the method. We do argue that quantum mechanics has been useful in physics for guaranteeing the stability of the atoms. Also quantum interference suggests that quantum probabilities can be canceled (while in classical statistical there is no analog method to cancel probabilities), and such cancellation seems like a positive aspect of quantum theory. However, we also say we do not examine the elastica solution to an empirical level and have not carried a demonstration that robustness will indeed be obtained for the quantum elastica. We hope future work will examine such possible advantages and possible disadvantages.  However, the scope of the work is to show that it is possible to consider the quantum method as a new statistical method and we show one way to do it. 

In the introduction of the elastica problem, the authors formulate it
in a Lagrangian way.  The path integral method is then introduced very
briefly and rather qualitatively, albeit this is the main apparatus
used in the present contribution. The authors essentially follow the
steps that can be understood from Feynman's original work (cited as
Ref. [13], albeit the citation is wrong, the correct volume of this
paper is 20 not 2), but even this is not stated.  As the quantum
information community dominantly uses Hamiltonian formulations, this
limits the accessibility of the description to those who are more
familiar with the Feynman path integral method, even though the
authors seem to address the optimization and quantum computation
community.

Thank you!, the citation has been fixed. We used the Feynman path integral method to derive the Quantum Elastica Equation and despite the brief introduction, the technique is fully developed for the Elastica. Yes, it is not so different than the type of calculations Feynman derived Schr\”oedinger equation, but also it is not exactly the same. In any case, we are happy to stress more to the reader that these calculations follow similar calculations by Feynman ….  

Not sure how the Hamiltonian formulation would have worked out (note that the obtained quantum Hamiltonian is not just the kinetic term coming straight from the classical kinecitc term (the Lagrangian is just one kinectic term). There is also a term that links the gradient of the position (the conjugate to x) of the elastica with the tangents. We have now pointed out this observation. Privately, a colleague that read the pre-paper asked a similar question and said that he would develop such derivation. This was a few weeks ago. We will then let this researcher do so or other researchers may do it. 

Section 2 then introduces the quantum elastica equations, and then
quantum mechanics enters, rather suddenly especially for those not
familiar with path integrals. But even so, why do we need the constant
hbar appear here, if once it is not supposed to be a physical problem,
just a physical method to solve it? Some explanation should have been given.

Thank you for the question. We created a new section in the paper, it is the new section 2 where there is a discussion of a need of an extra parameter in every statistical theory that is derived from a deterministic classical optimization theory. Moreover, we present how to obtain this parameter and that it depends on the domain of application. Since we have not yet applied this model to any domain problem, there is no empirical study, and we leave it as future work to estimate the parameter a quantum method must introduce. 

A more important problem is the introduction and implementation of the
wave functions. It is said that the wave function is in "Hilbert
space", but no details are given about this Hilbert space? The
consideration also lacks a proper description of the measurement process
which would be the basis of the probability interpretation. Without
the definition of measurement it is questionable whether the function
P_L(x,\theta) after Equation (17) obeys a probability
interpretation.

We clarify that the Hilbert space is comprised of function in x and \theta with the standard definition of Hilbert space for non-compact and compact  spaces.  We also clarify in the new section 2 that the conjugate of theta is curvature where the uncertainty principle applies, while x’s is the conjugate of the imaginary grad x and so the function is well defined in x, theta.

Regarding measurement, we also expanded the subject in the new section 2.  We see measurement as an empirical method of acquiring knowledge about the variable being measured. This is applicable for any domain, including (quantum) physics or (statistical) physics.  Then, the probabilities obtained reflect the possibility of the outcome of such variables when a measurement is made. It is special in quantum theory that conjugate variables cannot be measured simultaneously, so the acquisition of knowledge is never complete. Hopefully the reviewer will agree with the way we present measurements and the justification for the interpretation of  P_L(x,\theta). 

 Moreover, lemma 6 states that Psi^L satisfies the
quantum elastica equations, but Eq.(11) is for a function with a
variable s, so this cannot be even substituted into it.

We agree with reviewer. Thank you. We revised the paper and clarified the meaning of the quantity that is the product of the forward and backward solution. It is a solution that evolved for length L  and satisfy both boundary conditions, while the index s has a different interpretation.  This was part of the major revision we did.

A similar lack of interpretation appears at other points of the
manuscript, too. For instance, phase space entropy is introduced in
Definition 1, but its operational meaning is not mentioned. Meanwhile,
the argument around Eq.(19) suggests that a known stochastic treatment
can be recovered from the present formulation, which looks like a very
interesting point that would deserve more emphasis and discussion.

Section 2 The stochastic development is mentioned in the introduction and more emphasis is made to the value of the Wick rotation to bring a quantum statistical theory to a classical statistical theory. In the case of the elastica, David Mumford (a field medalist) have developed the statistical approach to this problem, and it is interesting that the statistical equation he arrived can be obtained as the Wick rotation to the quantum Hamiltonian we obtained. We are not sure how much more emphasis we can place than what we do in this newly revised paper.

Regarding the entropy: now that we discuss measurement in the new section 2, we emphasize it as knowledge acquisition and in a statistical theory entropy provides a quantification of the (lack of) knowledge of a probability distribution, as Shannon defined.  So the value of defining entropy becomes more clear as the definition of (lack of) knowledge.  Due to the uncertainty principle, the entropy must be in phase space and so the Fourier formulation of the problem in the appendix is tight to the entropy formula. The main motivation in the paper to use the concept of the entropy is to clarify that the forward solution disperses the initial state that is localized (thus we conjecture it increases this entropy) while the backward solution (when seen as a forward solution towards the end boundary condition) concentrates the probability distribution (conjectured to reduce the entropy). So, the entropy allows us to distinguish the behavior of the forward and the backward solution. Also, regarding the question about measurement, it is clear that the uncertainty is  between the space (x, theta) and their  Fourier variables, namely (m, k, n). This discussion now follows the entropy definition. 

Looking at the calculational details, the beginning of the proof of
Lemma 6 seems to be incorrect, when carrying out the substitution in
the unnumbered equation after equation (A13), a -((hbar^2)/4)
(\partial_\theta \Psi)(\partial_\theta \Psi^CPT) summand seems to be
missing from the right hand side. It should be at least doubly checked.

Yes, thank you. As mentioned above the solution Psi^L (as a function of s) is not the solution to the QEE, and the way to interpret it is as a function of $L$, i.e., it is the solution of the equation that evolved for length L and satisfies both boundary conditions.  Lemma 6 proves this statement.

Altogether the manuscript intends to give a long-term vision of a new
and original approach, yet with not yet justified benefits.  The calculations are technically involved, but the correctness of some statements is questionable, important interpretational details are missing, and the description is less accessible. Hence, I cannot, unfortunately, recommend its publication.

We thank the reviewer; the feedback was very valuable and we believe we addressed it and improved the paper due to it.

Reviewer 2 Report

Comments and Suggestions for Authors

1: The significance of the research needs to be highlighted better. The abstract and introduction don’t effectively convey the practical implications and potential benefits of the proposed method.  

2: The paper lacks one simple example of the application of this method. It would be beneficial if the authors added examples of the application of their technique.  

3: The need for a comprehensive baseline comparison suggests that broader method comparisons would illustrate the method's effectiveness. Please add a table and summarize the key advantages and disadvantages of the existing method.

4: Add some new references regarding the quantum elastica.

5: Could you please add some details regarding the effectiveness of the proposed method in future experiments and provide a simple example of how it can reduce time complexity?

Author Response

First, thank you so much for your feedback! We revised the paper and change the emphasis by tunning down any claim that we demonstrate that  quantum theory will be advantageous to classical statistical theory, we did not. We did not apply the quantum elastica to any empirical domain, so this study remains for the future. The claim is that we are proposing to use quantum theory to other domains, demonstrating how to do it, and extending the path integral method to scenarios with two boundary conditions, at the start and at the end of a path.  Again, we thank you very much for the feedback. 

1: The significance of the research needs to be highlighted better. The abstract and introduction don’t effectively convey the practical implications and potential benefits of the proposed method.  

Yes. The significance we demonstrate is conceptual, we mean that Quantum theory is a Statistical theory, and we show how to use quantum theory as a statistical theory in a problem that is not physics. We also showed that Wick rotation is a quantum mathematical tool that allows to transform quantum statistical theories to classical statistical theories, beyond physics.  As for comparing the two theories, we pointed out some differences between the theories such as the cancellation of probabilities exhibited in quantum theory (exemplified in the double slit experiments). Also, quantum theory, as a statistical theory has been very useful to physics to explain the stability of atoms. But we agree, we have not applied to any empirical domain and therefore we have not demonstrated that the use of quantum theory will lead to practical benefits beyond physics, only that will have different properties than classical statistical theory.  In our introduction we tuned down the benefits of quantum theory saying what we say above. 

Finally, as a contribution to the path integral method, we showed how it can be applied to two boundary conditions. This is a novelty to quantum theory, as a statical method.

2: The paper lacks one simple example of the application of this method. It would be beneficial if the authors added examples of the application of their technique.  

Yes, this is a fair point and limitation of the paper. We tried to point this out in the revised version, that we think the value of the paper is to show quantum theory as a statistical theory that can be applied to any field of optimization of an action.  Same point made on the answer above.

3: The need for a comprehensive baseline comparison suggests that broader method comparisons would illustrate the method's effectiveness. Please add a table and summarize the key advantages and disadvantages of the existing method.

As said above, we cannot claim that the method is better than statistical classical method. The differences of the two methods have ONLY been studied in physics where such advantages are known (and we point out: robustness /stability of the atoms are only explained with quantum statistical theory).   This paper is to show that quantum theory can be applied to an on-physical problem.

4: Add some new references regarding the quantum elastica.

Which reference you may suggest? There are so many different papers on it. We gave a brief history and pointed to papers that the reader can see the history of the elastica as well as references. Wehave been influenced by the work of Mumford that brought the classical statistical framework to it.  We mentioned it and showed how through Wick rotations we obtained his equation. We

5: Could you please add some details regarding the effectiveness of the proposed method in future experiments and provide a simple example of how it can reduce time complexity?

We have given more attention to the theory and measurements. We also made clear that deploying this material to an empirical field is the only way to estimate parameters, study the computational process. This is left as future work. Hopefully we (or someone else) will do such study. This  is not within the scope of the paper.

Reviewer 3 Report

Comments and Suggestions for Authors

See attached file.

Author Response

Thank you so much! We believe we fixed all of them. Regarding x versus (x,y) we just eliminated (x,y), everything is x now. 

thank you

Round 2

Reviewer 1 Report

Comments and Suggestions for Authors

The authors have thoroughly revised the manuscript and gave detailed and deep enough answers to the criticisms raised in my previous report. The mistakes I had pointed out have now been corrected and I did not find any others. Even though I'm still not fully convinced about the benefits of the presented ideas, I do admit that the idea is new and innovative, and it may possible be of interest in the light of possible future research. And indeed, in the revised version, this is exactly what the authors claim.  Some details of the interpretation could also be disputed, but this is normal when considering quantum mechanics, and the authors' standpoint is acceptable. In conclusion, the paper is now sound and not misleading, and it contains innovative ideas which can be of interest to the community. Hence, I recommend the publication of the paper as it is.

I recommend one very minor modification of grammatical nature though:

in manuscript line 45, there is :

"Studies show that classical simulating a quantum algorithm for certain problems,"

should rather read

"Studies show that the classical simulation  a quantum algorithm for certain problems,".

I recommend to correct it on proof.

Reviewer 2 Report

Comments and Suggestions for Authors

In general, the authors have done a good job of correcting the article according to the reviewer's comments. They have made significant changes to the text by providing more detailed explanations.